# A random optical parametric oscillator

Pedro Tovar [1] ✉, Jean Pierre von der Weid [2], Yuan Wang[1], Liang Chen[1] & Xiaoyi Bao[1]

Synchronously pumped optical parametric oscillators (OPOs) provide ultrafast light pulses at tuneable wavelengths. Their primary drawback is the need for precise cavity control (temperature and length), with flexibility issues such as fixed repetition rates and marginally tuneable pulse widths. Targeting a simpler and versatile OPO, we explore the inherent disorder of the refractive index in single-mode fibres realising the first random OPO – the parametric analogous of random lasers. This novel approach uses modulation instability ($\chi^{(3)}$ non-linearity) for parametric amplification and Rayleigh scattering for feedback. The pulsed system exhibits high inter-pulse coherence (coherence time of ~0.4 ms), offering adjustable repetition rates (16.6–2000 kHz) and pulse widths (0.69–47.9 ns). Moreover, it operates continuously without temperature control loops, resulting in a robust and flexible device, which would find direct application in LiDAR technology. This work sets the stage for future random OPOs using different parametric amplification mechanisms.

The increased interest in optical parametric oscillators (OPOs) in the past decade is driven by the need for high-power light sources with wide wavelength tuning, which are essential in applications such as trace gas detection[1–3] and spectroscopy[4–6]. Since the emission of OPOs is given in pairs of correlated photons, they are also of great demand as non-classical light sources for quantum optics applications[7,8]. The primary focus of recent research on OPOs has been the fabrication of compact resonators using microcavities with high quality factors[9–11]. However, there has been no progress in turning OPOs into more versatile and practical devices. OPOs usually comprise expensive pump lasers and an additional complex cavity including a highly non-linear component. Furthermore, the cavity must be precisely controlled to sustain parametric oscillation; typically, this is achieved by employing temperature-stabilised ovens[12], or by fine tuning the cavity length[13], practices feasible solely within undisturbed environments. The lack of a less complex and robust solution is hindering the widespread utilisation of OPOs.

The concept of random feedback, known for its fascinating properties and the requirement of minimal configuration, has been investigated in laser oscillators. Random lasers (RLs) consist of a special class of lasers with random feedback promoted by multiple scatterers. RL developed with optical fibres, the so-called random fibre lasers (RFLs), stand out in the class due to simple manufacturing and remarkable features. For instance, RFLs offer higher efficiency compared to dye or powder-based RL as light is confined to nearly one dimension[14,15]. The most common choice for random distributed feedback (DFB) in RFL is Rayleigh scattering[16,17], as it naturally occurs in single-mode fibres (SMFs) due to small refractive index fluctuations that arise during fabrication. Even though a large variety of optical gains were explored in the development of random lasers[18], including laser dyes, stimulated Raman scattering (SRS), stimulated Brillouin scattering (SBS), rare-earth ions, dye doped liquid crystals, and semiconductor optical amplifiers, a random optical oscillator with parametric amplification has never been achieved.

Two challenges precluded the development of a random OPO up to now. First, parametric conversion requires phase-matching for efficient amplification, which is inherently in contrast to the concept of a stochastic feedback. Second, in the case of synchronously pumped optical parametric oscillators (SPOPO), the repetition rate of pump pulses must match the cavity round-trip time; a requirement apparently unfeasible with open cavities defined by multiple scatterers. In fact, although many RL have been achieved with pulsed pumping, pulses were much longer than the cavity length, thus acting as a continuous wave (CW) pump. An exception was recently reported in ref. 19, where the cavity length was much longer than the pulse width. Authors made use of Rayleigh feedback in a short fibre section determined by the pulse width to achieve coherent emission. However,

[1]Nexus for Quantum Technologies, University of Ottawa, 25 Templeton Street, Ottawa, ON K1N 6N5, Canada. [2]Centre for Telecommunication Studies, Pontifical Catholic University of Rio de Janeiro, 22451-900 Rio de Janeiro, RJ, Brazil. ✉e-mail: ptovarbr@uottawa.ca

a double-gain configuration had to be used to compensate the large losses in the cavity.

Here, we demonstrate the first random optical parametric oscillator (R-OPO), in which parametric amplification is provided by modulation instability through the $\chi^{(3)}$ non-linearity in a single-mode fibre, and feedback is from Rayleigh backscattering in a novel piecewise distributed half-open cavity. We explain the physical principle behind parametric oscillation with random feedback, as well as characterise the main features of the R-OPO, including its threshold power, saturation, wavelength stability and coherence time. As we shall see, the R-OPO turns to be a highly coherent and versatile light source, allowing tuneability of (1) the pulse repetition rate; (2) the pulse width; and (3) the emission wavelength, all in a robust and simpler configuration compared to typical OPOs.

## Results

### Principle of operation

Modulation instability (MI) is a parametric conversion process that comes from an interplay between the Kerr effect and dispersion. It offers parametric amplification, which can be even higher than that obtained from Er-doped fibre amplifiers (EDFA) or Raman amplifiers, reaching up to 70 dB in some cases[20]. For the efficient generation of MI gain, other non-linear effects with low threshold, such as SBS and SRS, should be suppressed. Pulses shorter than ~10 ns were shown to sufficiently mitigate SBS/SRS effects and efficiently promote MI gain[21], which enables oscillation in the R-OPO as described below.

The R-OPO operation principle is illustrated in Fig. 1. Short duration pump pulses at high repetition rates are launched into an SMF through a fibre Bragg grating (FBG) with Bragg wavelength $\lambda_B$ and bandwidth $\Delta\lambda$. The central wavelength of pump pulses, $\lambda_0$, is selected to lie outside of the FBG reflection bandwidth, so that the FBG is transparent to $\lambda_0$. As a pulse propagates into the fibre, MI sidebands are generated and co-propagate with the pulse, offering wide band optical gain[22]. The Bragg wavelength $\lambda_B$ is set to match one of the wavelengths of peak MI gain (see inset in Fig. 1), such that the Rayleigh backscattering radiation from MI sidebands in the vicinity of $\lambda_B$ is reflected at the FBG and re-injected into the fibre. Thus, a half-open cavity is formed for lightwaves with wavelength within $\Delta\lambda$, i.e., photons are partially trapped between a fixed Bragg reflection and random DFB.

Half-open cavities similar to the one in this study were used before in the development of random DFB fibre lasers with SBS gain[23,24]. However, those cavities were pumped with CW light, so that the Rayleigh backscattered light from the whole fibre was reflected at the FBG, re-injected into the cavity, and amplified. This is not the case when the cavity is pumped with pulsed light. In this scenario, the Rayleigh

backscattered light will only undergo MI amplification if it arrives at the FBG at the same time as a new incoming pulse[19]. This only occurs for light backscattered at specific fibre sections in phase with pump pulses, which are defined from the pulse repetition rate and pulse width. We refer to these sections as effective Rayleigh sections, indicated in Fig. 1, since the backscattered light exclusively from these sections oscillates coherently in the cavity. Therefore, this new type of cavity can be viewed as a piecewise distributed cavity, where feedback is at the same time randomly distributed and discontinuous.

For an SMF of length $L$, there is an integer number of effective Rayleigh sections, with length depending on the pulse width, and positions defined by the pulse repetition rate, $f$. Positions can be calculated from $z_m = m \cdot v_g \cdot (2f)^{-1}$, where $v_g$ is the group velocity, and $m = 1, 2, \ldots, M$, with $M$ being the total number of fibre sections equal to $\left\lfloor \frac{L}{v_g/2f} \right\rfloor$. By increasing the repetition rate, the number of effective Rayleigh sections also increases for a fixed fibre length, and as a result the cavity's reflectivity is increased since it is given by the superposition of Rayleigh light from all sections. Hence, by simply tuning the repetition rate one can accurately calibrate the reflectivity, thus balancing the cavity losses with MI gain and ultimately enabling random parametric oscillation. The total reflectivity of the cavity can be derived from the theory of optical time-domain reflectometry (OTDR)[25], and it is given by the summation of light reflected in each section $m$, $R_m$, expressed by

$$R_{\text{tot}}(f) = \sum_{m=1}^{M} R_m(f) = \sum_{m=1}^{M} S \cdot \frac{\alpha_s}{\alpha} \cdot e^{-2\alpha z_m(f)} \left(1 - e^{-\alpha W}\right), \quad (1)$$

where $S$ is the Rayleigh capturing coefficient, $W$ is the pulse width, $\alpha$ is the total loss coefficient and $\alpha_s$ is the scattering coefficient.

The parametric oscillation threshold occurs when the MI gain compensates the cavity losses, which can be calculated through

$$\sum_{m=1}^{M} g_p(z_m, P_0) \left(\xi R_m(f)\right) R_{FBG} > 1. \quad (2)$$

In the equation above, $g_p$ is the position- and power-dependent peak MI gain[26], $P_0$ is the input power, and $R_{FGB}$ is the FBG's reflectivity. As described in Eq. (1), $R_{\text{tot}}$ corresponds to the power reflectivity of fibre, or the mean reflection loss. Were the backscattering optical field from all effective Rayleigh sections added coherently, then coherence spikes would be randomly present in the reflectivity spectrum[27,28], similar to the spectral reflectivity signature of RFGs[29]. The constructive interference of backscattering light produces sharp reflectivity peaks

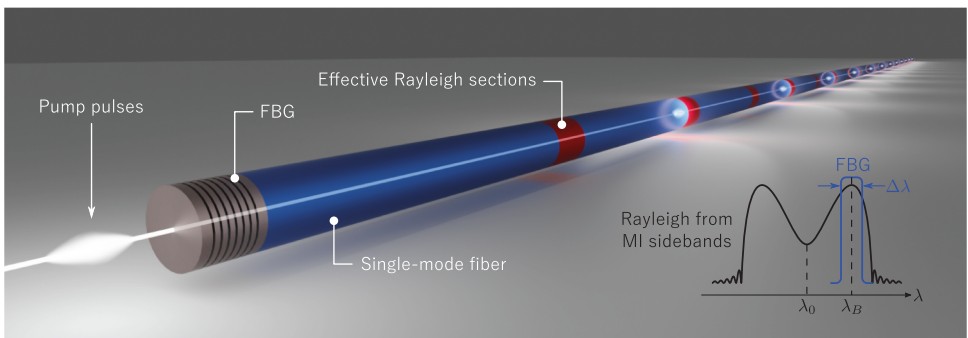

**Fig. 1 | Principle of random optical parametric oscillation.** Pump pulses at $\lambda_0$ with high repetition-rate are launched through a fibre Bragg grating (FBG) into a long single-mode fibre, where modulation instability initiates when the pulses carry sufficient energy. Rayleigh backscattering from repetition-rate-matched zones along the fibre (effective Rayleigh sections shown in red) meet an incoming pump pulse when reflected at the FBG, thus undergoing parametric amplification while co-propagating with the pump pulse. An illustration of modulation instability backscattered spectrum is shown on the right, with the FBG central wavelength tuned to select one of the spectral regions of highest parametric gain, thus forming a piecewise distributed random cavity for lightwaves with wavelength within the FBG bandwidth, enabling random parametric oscillation.

at certain wavelengths, depending on the position of the effective Rayleigh sections. The peak values exceed the mean reflection by, typically, a factor 10 (see calculations in Supplementary Notes and Supplementary Fig. 1). This coherence factor is indicated as $\xi$ in Eq. (2), and it must be considered in the calculation of the oscillation threshold since parametric oscillation would preferably start at the wavelength of highest reflectivity peak within $\Delta\lambda$.

## Onset of random OPO

The experimental validation of the R-OPO was performed as follows. We conducted two sets of experiments, with and without an FBG in the setup. In both cases, pump pulses with 5 ns duration and peak power tuned within the range 250 to 600 mW were launched into a 5.25 km-long SMF. Pulses were prepared from a semiconductor laser at 1549.05 nm and two electro-optic modulators (see Fig. 2a), so that the repetition rate could be readily tuned. Pulses were initially modulated at 1 MHz and amplified before injected into the fibre. The optical spectrum of transmitted signal at the end of the fibre was analysed (point $A$ in Fig. 2a), and the rise of symmetrical MI sidebands was observed for the case without the FBG (Fig. 2b). When launching the pulses through the FBG ($\lambda_B = 1550$ nm and $\Delta\lambda = 0.28$ nm), a half-open cavity is formed; Rayleigh backscattered light will be reflected back into the fibre and oscillation will occur if sufficient MI gain is available. At an input peak power of 345 mW, a sharp line is observed at the FBG wavelength, indicating the onset of random parametric oscillation. This is shown in Fig. 2c, which also displays the reflection spectrum of the FBG (grey curve). Once parametric oscillation initiates, four-wave mixing (FWM) processes start immediately from the interaction between the pump light and the R-OPO emission, giving rise to multiple FWM products at wavelengths satisfying the phase-matching condition. This mixing produces the comb-like profile in the full spectrum that would be perceived in time domain as a high frequency pulse train within the output pulse envelope. Indeed, this is only possible because the fibre dispersion is low (4.7 ps/nm·km) and the walk-off between FWM frequencies is much smaller than the pulse length.

The theory discussed in the previous section predicting the threshold for random parametric oscillation was verified by measuring the reflectivity of the piecewise distributed Rayleigh scattering, $R_{tot}(f)$ (experiment details in Supplementary Methods; see experimental setup in Supplementary Fig. 2). Figure 3a shows the measured and theoretical reflectivity as a function of the pulse repetition rate when launching 5 ns pulses into the SMF. The fibre reflectivity is an ever-increasing function of the repetition rate, with two effects contributing to the increase. First, any small increment on the repetition rate pulls all effective Rayleigh sections closer to the launching end, so that light experiences less attenuation when propagating towards and backwards from each section. The second cause is the inclusion of a new effective Rayleigh section at the end of the fibre, which translates into a sudden jump in reflectivity. These jumps become more evident when analysing the result in a smaller frequency scale from 100–200 kHz (Fig. 3b). By calculating the number of pulses (or Rayleigh sections $M$) in the fibre as a function of repetition rate, as shown in blue circles for groups of data points sharing the same number of pulses in the fibre, we clearly verify the contribution of each added section, which is accurately predicted through the use of Eq. (1).

The threshold power was determined by varying the pump power and measuring the power of the R-OPO oscillating line around 1550 nm at the output point $A$ (see Fig. 2a). The experimental result is shown in Fig. 3c. A threshold power is noted at 337.5 mW for the repetition rate of 1 MHz, together with a steep linewidth reduction, indicating the establishment of a coherent process. The efficiency of the system is found to be of $2.6 \times 10^{-3}$, resulting from the weak feedback of Rayleigh scattering zones along the SMF, and the high launching power required for MI gain. Since the number and location of the effective Rayleigh sections depend on the frequency of pump pulses, the R-OPO threshold was measured as a function of the repetition rate. This measurement is displayed in Fig. 3d, where the threshold power does not steadily decrease with the repetition rate, displaying rather a rippling behaviour. The theoretical calculation from Eq. (2) actually predicts an oscillatory pattern for the R-OPO threshold power, which comes from an interplay between the distributed MI gain and the reflectivity. When sections are pulled closer to the launching end with the increase of the repetition rate, each section experiences less MI gain since the gain is length-dependent. Thus, even though the reflectivity increases, there is less gain for each section, so more power is required to reach the threshold. Eventually, the increase of the repetition rate will result in the addition of a new effective Rayleigh section at the end of the fibre, which will experience full gain along the whole fibre, resulting in an abrupt decrease in threshold power.

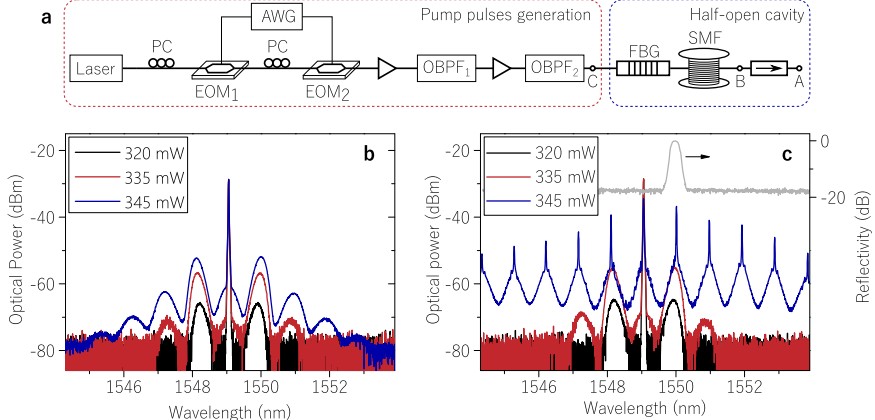

**Fig. 2 | Experimental validation of random OPO. a** Experimental setup of the random OPO: high-extinction-ratio pump pulses were prepared from a semiconductor laser at 1549.05 nm and a combination of two electro-optic modulators (EOM) driven by an arbitrary waveform generator (AWG). Polarisation controllers (PC) were used for maximal transmission at the EOMs. Pulse peak power was controlled through two amplification stages including Er-doped fibre amplifiers and optical band-pass filters (OBPF), the latter used to minimise amplified spontaneous emission noise. The half-open cavity where random parametric oscillation takes place is composed of a fibre Bragg grating (FBG) and a 5.25 km-long single-mode fibre (SMF). An optical isolator at the end of the SMF prevents unwanted point-reflections. Optical spectra were collected at the output of the isolator (point $A$) at three input powers without and with the FBG in the setup, as shown in (**b, c**), respectively. Above the parametric oscillation threshold, a narrow laser-like line rise on top of the modulation-instability sideband at which the FBG is centred. Grey curve displays the FBG reflectivity.

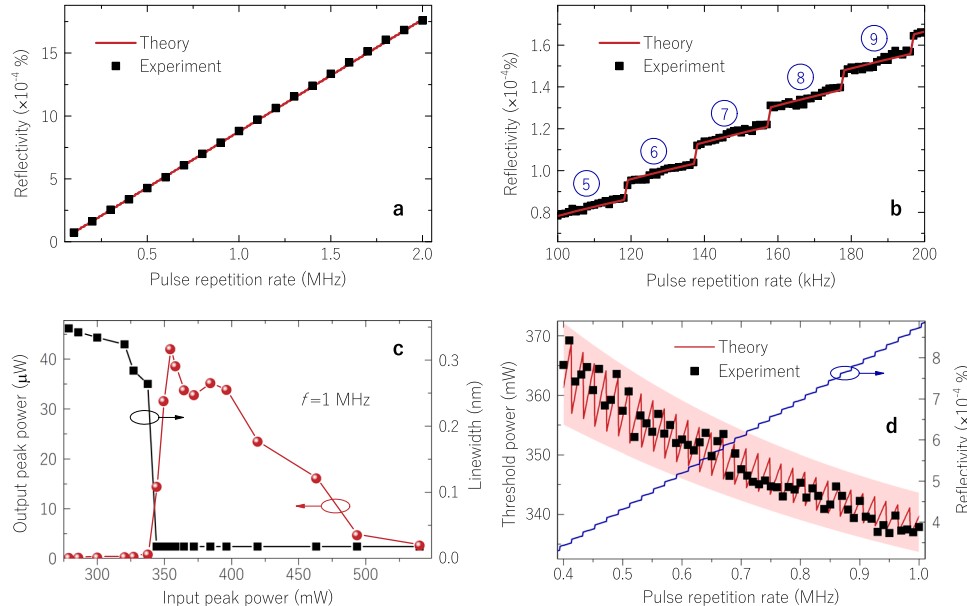

**Fig. 3 | Reflectivity characterisation and oscillation threshold. a** Reflectivity of piecewise distributed Rayleigh scattering in a single-mode fibre as a function of the pulse repetition rate. **b** Measuring the reflectivity for a narrow span of pulse repetition rate, 100–200 kHz, indicates reflectivity discontinuities at repetition rates corresponding to the addition of a new reflecting section at the end of the fibre; blue numbers indicate groups of data points sharing the same number of reflecting sections. **c** R-OPO output peak power and linewidth as a function of input peak power for a repetition rate of 1 MHz; and **d** the threshold power dependence on the repetition rate. The threshold power decreases with repetition rate due to the increase of cavity's reflectivity, which translates into a decrease in cavity losses. The blue curve is the theoretical reflectivity of piecewise distributed Rayleigh scattering.

Although the theoretical calculation provides a good estimation for the experimental threshold, there is an uncertainty element that precludes finer accuracy. At the threshold, parametric gain slightly overcomes the losses at a certain wavelength within $\Delta\lambda$ where a constructive interference occurs from light backscattered at multiple effective Rayleigh sections. An incremental increase of the pulse rate completely changes the positions of effective Rayleigh sections, which could lead to a stronger or weaker interference peak. The coherence factor $\xi = 10$ used in theoretical calculations is an average value, but deviations of about ten percent are expected when changing the positions of effective Rayleigh sections (see Supplementary Notes). The upper and lower bound envelopes considering a deviation of ten percent of $\xi$ are shown in Fig. 3d, where a light-red threshold band is shown, giving a solid confidence interval for threshold calculations. Another interpretation is given in the framework of phase-OTDR systems. Coherent Rayleigh backscattering in phase-OTDR traces is known from its jagged profile owing to the interference of light within the pulse width. By arbitrarily selecting $m$ fibre sections equally spaced from the phase-OTDR trace and adding them together, the result can be either a high or low intensity signal. Re-selecting new positions slightly shifted from the previous ones would randomly affect the result, as is the case for the R-OPO's threshold.

It must be highlighted that the tuneability of the pulse repetition rate is a feature exclusive of R-OPOs and not allowed in other SPOPOs[21,30,31]. Here, any repetition rate supports oscillation, being evidence of parametric oscillation from a random distributed feedback, and putting the R-OPO apart in a new class of synchronously pumped parametric oscillators with tuneable repetition rate.

Although SBS and SRS could be efficiently suppressed by launching short pump pulses, undesired non-linear phenomena would be present if the power is sufficiently high. The R-OPO peak power rises linearly when increasing the input power from 337 to 354 mW (Fig. 3c), but further increasing the pump power leads to the build up of self-phase modulation (SPM), which grows at the expense of MI gain, leading to the saturation of output power. The linear operating range is increased at higher repetition rates: given the lower threshold obtained at higher rates, the SPM-free regime is naturally enlarged, adding flexibility to the system. On the other hand, the linear operating range is reduced at lower repetition rates. In fact, the MI gain depletion limits the minimum repetition rate at which parametric oscillation can be sustained for a given pulse width; we found that limit at ~0.4 MHz for a pulse width of 5 ns (Fig. 3d). Nonetheless, as we shall see in the next section, by enlarging the pulse width and hence increasing the backscattering power, random parametric oscillation can be achieved with repetition rates much smaller than 0.4 MHz.

## Time-domain analysis

Unlike random DFB fibre lasers, which emit stochastic pulses near the threshold[16,27], R-OPO pulses are periodic and well-defined. The shape of output pump and R-OPO pulses, respectively at 1549.05 and ~1550.00 nm, was measured by filtering the corresponding wavelengths at the output (point *A* in Fig. 2a), and the result is shown in Fig. 4a, b. It is clear that R-OPO pulses are shorter than pump pulses; strong parametric interaction, and hence large parametric gain, is temporally confined to the peak/flat-top portion of the pump pulse. Pump photons in this portion of the pulse are largely annihilated to enable parametric oscillation, depleting the centre of the output pump pulse. R-OPO pulses build up with a duration mostly defined by the flat-top portion of pump pulses, yielding a duration of ~2.2 ns.

In light of the rising interest in developing OPOs with tuneable pulse widths[32], we conducted a controlled experiment varying the pump pulse width to investigate its impact on random parametric oscillation. In principle, since less light is backscattered by shorter effective Rayleigh sections (shorter pulses), more sections (higher repetition rate) are needed to reach the oscillation threshold for the same pump power. This effect was experimentally observed, as higher repetition rates had to be set in order to achieve random parametric oscillation for shorter pulses. Yet, by increasing the pump power, a large combination of repetition rates and pulse widths could be obtained. Again, measurements were performed by filtering the R-OPO's output (point *A* in Fig. 2a) at the oscillating wavelength (~1550 nm), and the result is displayed in Fig. 4c. The duration of pump pulses

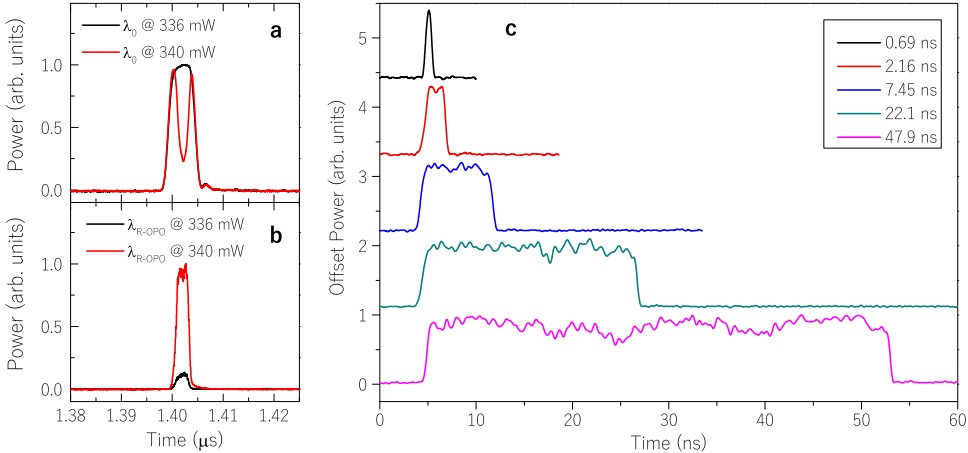

**Fig. 4 | R-OPO pulses shape and tuneable pulse width. a** Pump pulses measured at the fibre output below (336 mW) and above (340 mW) the oscillation threshold. The start of parametric oscillation is identified by a strong depletion in the centre of pump pulses. Pulses at the R-OPO wavelength are shown for the same pumping conditions in (**b**), where a significant power increase is seen above the oscillation threshold. **c** By varying the pump pulses duration from 2 to 50 ns, R-OPO pulses were realised with durations from 0.69 to 47.9 ns.

was set to 2, 5, 10, 25 and 50 ns, resulting in R-OPO pulses with widths 0.69, 2.16, 7.45, 22.1 and 47.9 ns, respectively. For all pulse widths tested, the repetition rate could be tuned from 700 kHz up to 2 MHz. For pulse durations longer than 10 ns, much lower repetition rates could be achieved, down to 40 kHz. Since wider pulses define longer effective Rayleigh sections, pulse durations much longer than 50 ns at repetition rates higher than 2 MHz would nearly correspond to a CW operation as effective Rayleigh sections would be too close to one another, which is the reason why we limited the width of pump pulses to 50 ns. Not surprisingly, longer pulses resulted in smaller threshold powers, offering a more flexible operation range in terms of pump power, not much limited by SPM as observed in Fig. 3c for 5 ns pulses. By adding an optical circulator to the setup in port $C$ of Fig. 2a, we monitored the backscattered spectrum and observed the rising of SBS for pulses larger than 10 ns. Even though SBS is no longer efficiently mitigated for wider pulses, the presence of SBS does not prevent random parametric oscillation here.

Removing the optical filter from the fibre output, we analysed the existence of ultra-fast pulses as predicted from the spectral results shown in Fig. 2c. The autocorrelation trace measured at an input power below the R-OPO threshold reveals few MI-induced oscillations with durations of 2.37 ps (Fig. 5a), agreeing with the result from the inverse fast Fourier-transform (IFFT) of the optical spectrum (inset of Fig. 5a). Above threshold, narrow pulses are measured (Fig. 5b) with a duration of 1.35 ps, also in agreement with the IFFT result from the comb-like spectrum (inset of Fig. 5b). As the autocorrelation measurement is an average result from a large number of autocorrelated pulse trains, hops of the R-OPO wavelength (analysed next) distorts the width of pulses non-centred in the autocorrelation trace ($t = 0$) as observed in Fig. 5b. The generation of narrow pulses is only possible because of the set of phase matched FWM by-products, which build up instantaneously with parametric oscillation covering a large wavelength range. There is a stringent phase-relation between the multiple high-order FWM products (signals and idlers in conjugate phase), thus contributing to the sharpening of ultra-fast oscillations and to the generation of the picosecond-pulses train.

## Spectral analysis
Next, we verified the stability of the R-OPO emission. As Rayleigh scattering is sensitive to temperature and strain variations with strong wavelength dependence, small environmental perturbations affect the interference of backscattered Rayleigh light, which may convert the

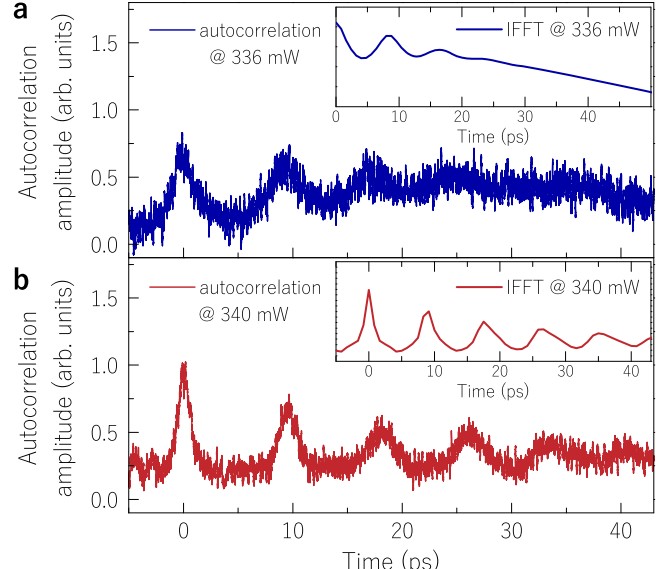

**Fig. 5 | Autocorrelation results. a, b** show the autocorrelation traces when measuring the output of the half-open cavity for input powers below and above the threshold, respectively. A sharp pulse train is observed for the latter case. Insets show the inverse fast-Fourier transform (IFFT) of corresponding frequency-domain optical spectra.

spectral reflectivity peak where the R-OPO is settled, say at $\lambda_1$, into a trough. However, a constructive interference, i.e., a coherence spike, will be found at another wavelength $\lambda_2$. In that case, parametric oscillation immediately resettles at $\lambda_2$. By tracing a parallel with random fibre lasers, the wavelength hops can be seen as a competition among a large number of longitudinal modes, and every time a new mode wins the competition a wavelength hop occurs. We monitored the R-OPO output over 2 min while pumping with 5 ns pulses and at a repetition rate of 1 MHz. The result is shown in Fig. 6a, b. It is clear that, even though no cavity control was employed, the R-OPO intensity is steady, not disturbed by the small wavelength hops by more than 13%, showing a standard deviation of 5%. This is an important step toward highly stable OPOs, as parametric oscillation here is robust against perturbations in the half-open cavity. In addition, even in the absence of a temperature

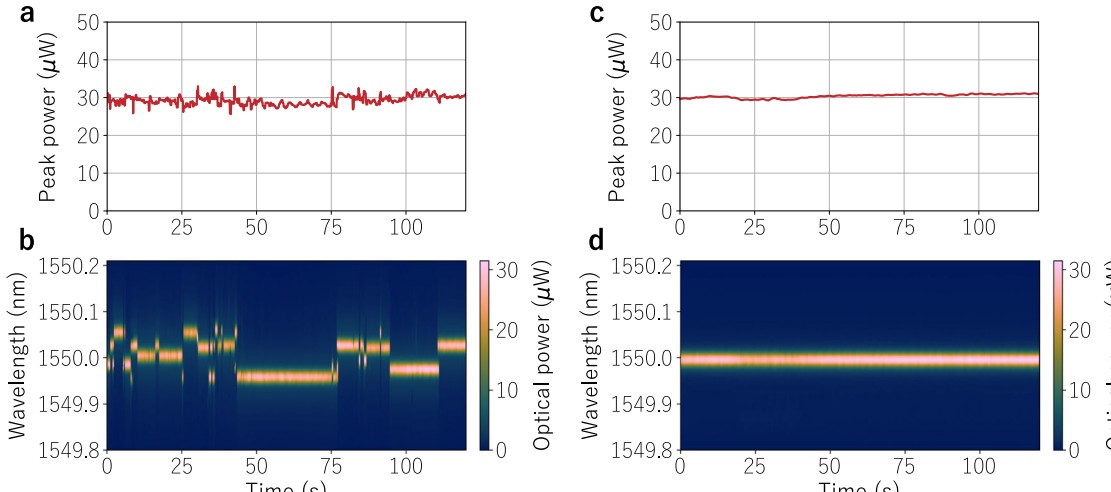

**Fig. 6 | Power and spectral stability. a** Analysis of the optical power of the R-OPO oscillating line around 1550 nm shows uninterrupted operation over two minutes at a repetition rate of 1 MHz and with a pulse width of 5 ns. The corresponding spectrum pattern shown in (**b**) indicates wavelength hopping in a scale of seconds. By adding an enhanced-Rayleigh fibre to the setup and reducing the pulse repetition rate to 19.6 kHz so that the number of effective Rayleigh sections is reduced to one and located at the added fibre, a wavelength hop-free operation is achieved (**d**). In addition, the output power was stabilised even further as shown in (**c**).

or vibration control loop, the emission remains consistently at the same wavelength for several seconds, occasionally exceeding 25 s. This duration is much longer than that in random DFB lasers, which exhibit wavelength drifts in the scale of milliseconds[24,33,34].

Even though the wavelength hops observed in Fig. 6b scarcely affect the output power, they result in a net-broadening of the long-term linewidth, which may be problematic from an application point of view. These hops come from the fluctuation on the interference pattern of light backscattered from all effective Rayleigh sections. Although the interference of light backscattered along a small fibre section is sufficiently stable (robust against small environmental perturbations), which is the reason why phase-OTDR works as a reliable distributed temperature/strain sensor[35], the stability is compromised when considering tens of sections separated by about a hundred metres each. A stable phase difference between interfering light is the main requirement for a steady interference pattern. In a small fibre section of length $d$, the phase difference between light backscattered at the beginning and at the end of the section is given by $\Delta\Phi = n(T, \varepsilon) kd$, where $k$ is the wavenumber, and $n$ is the temperature- and strain-dependent refractive index along $d$. Since $d$ is small (0.5 m), tiny variations of the refractive index do not represent big phase changes, so that the interference of backscattered light is stable. Now, let us consider the interference of light backscattered at two short fibre sections separated by a large distance $D$ (100 m). In this case, tiny index fluctuations are magnified by $D$ and may cause significant phase differences, thus directly affecting the interference pattern. Clearly, this effect is intensified when increasing the number of sections and the distance between sections; the result is a fluctuation of the effective reflectivity spectrum. Therefore, the spectral location of the highest reflectivity peak sustaining parametric oscillation fluctuates with environmental noise, causing the observed wavelength hops.

A solution to this problem is to reduce the number of effective Rayleigh sections to one. Obviously, this comes with the penalty of higher loss, since more sections translates into more backscattered power and a lower threshold for parametric oscillation. To mitigate this issue we cascaded the 5.25 km SMF with a 1 km-long enhanced-Rayleigh fibre (inserted in point $B$ of Fig. 2a), which provides about 16 dB extra backscattering light compared to standard SMF[36]. Therefore, a single section of the enhanced-Rayleigh fibre backscatters as much power as ~40 sections of SMF, but with a stable backscattered interference spectrum. Note that, the 5.25 km fibre is still required for high

MI gain, while the feedback comes exclusively from a single section (0.5 m) located at the enhanced-Rayleigh fibre. By setting the repetition rate to 19.6 kHz, we addressed a single section of 0.5 at 5272 m, i.e., in the beginning of the enhanced-Rayleigh fibre. We measured an emission spectrum similar to that shown in Fig. 2c, indicating the onset of random parametric oscillation. When monitoring the R-OPO oscillation line around 1550 nm over two minutes, wavelength hops were completely removed, as shown in Fig. 6d. Another positive outcome is observed in the peak power stability (Fig. 6c), which exhibits a steadier intensity, with a standard deviation of only 1.7%. There is, however, a new limitation in terms of repetition rate tuneability: since the enhanced-Rayleigh fibre has only 1 km, to locate the effective Rayleigh section along this fibre, we are limited to repetition rates in the range 16.6–19.6 kHz. Longer enhanced-Rayleigh fibres would immediately extend this range.

It is worth mentioning that the monitored optical spectrum shown in Fig. 6d was collected with the enhanced-Rayleigh fibre laying on top of an optical table without temperature/vibration control, thus susceptible to vibration noise from lab equipment and air-temperature variations. Clearly, as the temperature/strain in the addressed section significantly changes, a wavelength shift would be observed. This can be either solved through further stabilisation of the enhanced-Rayleigh fibre (e.g., put in a thermally isolated sound-proof box), or explored as a temperature/strain sensor with high spatial resolution and high signal-to-noise ratio (SNR). A high SNR is expected because one would measure the wavelength drift of a strong oscillating light, not from a weak backscattered light as in most fibre-sensors, acting similar to a laser-sensor[37,38].

## Coherence

Last, we investigate the coherence of R-OPO pulses. The interest in coherent optical pulses and their characterisation dates back from the 1970s[39,40], when interferometric measurements were performed to study the interference of subsequent pulses in a pulse train, i.e., inter-pulse coherence (or mutual-pulse coherence). We start by filtering the R-OPO oscillating line at 1550 nm with a band-pass filter at point $A$ of the setup in Fig. 2a. As in previous measurements of inter-pulse coherence, R-OPO pulses were sent to a Mach–Zehnder (MZ) interferometer as shown in Fig. 7a. For coherence characterisation, it is required that one of the interferometer's paths is delayed by the pulse repetition period so that two subsequent pulses temporally and spatially overlap. This is an easier task when the pulse repetition rate is

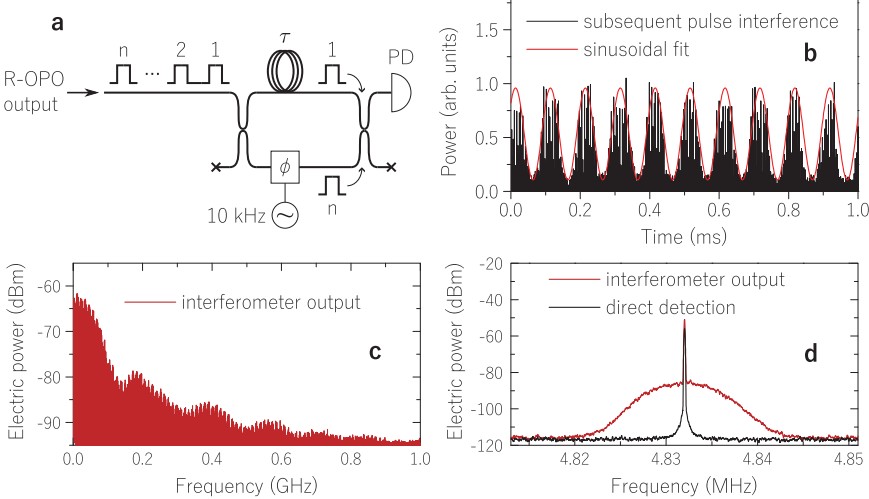

**Fig. 7 | Coherence measurements. a** Mach–Zehnder interferometer used in coherence characterisation measurements. **b** Setting a fibre delay of 167 m between the two paths of the interferometer and applying a phase modulation of 10 kHz to one of the paths, a strong interference pattern was detected by a photodiode (PD) at the output of the interferometer, indicating a high coherence degree between subsequent pulses. **c** Increasing the delay to 100 km to promote an incoherent beating while still ensuring the overlap between pulses from both paths, the PD's output was measured at an electrical spectrum analyser. The envelope is defined by the pulse spectrum, while narrow spectral lines, separated by the pulse repetition rate, are also observed. One of these lines, centred at 4.832 MHz, is investigated in more details as shown in (**d**), revealing a spectral width of 5 kHz, which corresponds to an optical linewidth of 2.5 kHz. When opening either of the interferometer's paths, the interference is killed and the spectrum reveals just the direct detection of optical pulses, showing only the electrical tones defined by the pulse repetition rate.

higher than hundreds of MHz, thus requiring a delay between the paths lower than a few metres[41,42]. Here, as we are working with repetition rates close to 1 MHz, a fibre delay $\tau$ of around 200 m is required. With a pulse length of about 44 cm (or 2.2 ns), it can be hard to impose an exact fibre delay so that subsequent pulses overlap. Fortunately, given the R-OPO tuneability of repetition rate, we can use any available fibre spool with approximately 200 m, detect the interferometer output with a photodetector (PD) and an oscilloscope, and fine tune the pulse repetition rate until pulses from both paths overlap in time. We ended up using an available fibre delay of 167 m and a pulse rate of 966,990 Hz. If subsequent pulses do interfere (coherent regime), by sinusoidally modulating the optical phase at one path of the interferometer with a low frequency, then the envelope of the interferometer's output would be intensity modulated, corresponding to constructive and destructive interference of subsequent pulses. We applied a phase modulation of 10 kHz to the bottom arm of the interferometer and the detected signal is shown in Fig. 7b, clearly exhibiting a 10 kHz sinusoidal modulation with a visibility higher than 80% (limited by the unbalanced intensities in the MZ paths), thus demonstrating that the R-OPO exhibits inter-pulse coherence.

For further coherence characterisation, we measured the coherence time as follows. We replaced the delay fibre with another spanning a distance of 100 km ($\tau = 0.5$ ms). The primary objective was to induce an incoherent beating at the output of the interferometer, akin to the principle of the delayed self-heterodyne method, but in pulsed regime. Again, the repetition rate was fine tuned so that pulses from both paths overlap in time. In this case, with a delay of 100 km and a repetition rate of ~1 MHz, the interferometer's output corresponds to the overlapping of pulse #1 with pulse #500. The electrical spectrum of the output is shown in Fig. 7c, whose envelope is defined by the pulse spectrum, but exhibiting narrow spectral lines separated by the pulse repetition rate. By zooming in any of these lines, for instance, the one centred at 4.832 MHz shown in Fig. 7d, we observe the incoherent beating between optical modes separated by 4.832 MHz, yielding a 3 dB-linewidth of 2.5 kHz, or a coherence time of ~0.4 ms. When any of the interferometer's paths is opened, or the repetition rate is mistuned (pulses not overlapping), we rather have just the direct detection of a pulse train (black curve in Fig. 7d), which simply shows narrow spectral

lines separated by the repetition rate with width defined by the stability of the electrical pulse generator (see AWG in Fig. 2a).

Such a long coherence time is attributed to a fundamental requirement for light oscillation. The onset of random parametric oscillation requires that light reflected at every effective Rayleigh section is in phase with each other, otherwise oscillation would cease after one round-trip. This strict requirement, guarantees that backscattering light from two effective Rayleigh sections apart from a few kilometres are in phase, which is preserved by MI amplification in the next round-trip, thus yielding a highly coherent pulsed light source.

## Discussion

This work demonstrates the first random OPO, which was based on modulation instability and Rayleigh scattering in single-mode fibres. Taking advantage of Rayleigh scattering as random distributed feedback mechanism, parametric oscillation is made possible at repetition rates ranging from 16.6 kHz up to 2 MHz, making the R-OPO a unique SPOPOs with tuneable repetition rate. It follows that the R-OPO emission is robust against environmental noises, thus offering continuous parametric oscillation. This represents an important step toward the development of highly stable OPOs, where any kind of temperature/vibration control is dispensed for continuous operation, however exhibiting wavelength hopping in a scale of seconds. For wavelength hop-free operation, we exploited the backscattering from an enhanced-Rayleigh fibre, which additionally demonstrated superior stability in output power.

Beyond offering repetition rate tuneability, the R-OPO pulse width could also be tuned from 0.69 ns up to 47.9 ns. Furthermore, we showed that the R-OPO presents inter-pulse coherence, having an ultra-long coherence time of ~0.4 ms. These three features, i.e., tuneable repetition rate, tuneable pulse width and long coherence time, make the R-OPO of special interest for LiDAR applications. A tuneable repetition rate is desired in LiDAR technology as it defines the sampling frequency of the target and influences on the measurement accuracy[43,44]. Compared to other light sources designed for LiDAR, for instance the work reported in ref. 43, where a Q-switched nanosecond laser was developed offering repetition rate tuneability from 1–4 kHz, the R-OPO offers significantly wider repetition rate tuning range.

Control of the pulse width is required in LiDAR as it enables manipulating both the pulse spectral content and the ranging efficiency[45], with the R-OPO exceeding the pulse width tuning range of other parametric light sources; e.g., in ref. 32, a doubly resonant OPO with tuneable pulse duration from 5.7 to 7.9 ns (signal) and 7.1–19.7 ns (idler) was reported. Coherent LiDAR outperforms the incoherent alternative as it enables the full recovery of phase, frequency and amplitude information[46]. In addition, since a coherent detection scheme is used in coherent LiDAR, a much lower pulse power can be used, translating into a more efficient system. Therefore, from its exceptional tuning flexibility and high coherence, it is expected that the R-OPO would find direct application in LiDAR technology.

Different from conventional OPOs, offering ultra-wide wavelength tuning, the R-OPO wavelength tuneability explored in this work is limited. Even though the FBG central wavelength can be tuned over several nanometres, the spectral location of MI sidebands is relatively fixed, depending mostly on fibre properties and on the pump power[47]. Changing the pump power only offers small wavelength tuning as can be seen from the shift of peak MI gain in Fig. 2b when the pump power is increased, and replacing the fibre is a non-practical solution. However, we experimentally verified that higher order MI sidebands can be explored to achieve random parametric oscillation, thus enabling wavelength tuning. By setting the central wavelength of the FBG to the −3rd or +3rd order MI sideband, the emission wavelength could be tuned in a range of 6 nm. Even so, the spectral tuneability remains significantly inferior compared to that of conventional OPOs. For instance, an OPO also making use of optical fibre in a hybrid fibre/free-space configuration allowed wavelength tuning in the range 1051–1700 nm[48], where tuning was accomplished by both varying the grating period of a periodically poled crystal and adjustment of the cavity length. Nevertheless, the concept of random parametric oscillation goes far beyond the setup explored in this work. Different parametric amplification mechanisms other than MI could be used for parametric gain. For instance, by exploring the $\chi^{(2)}$ non-linearity of crystals for parametric amplification, and Rayleigh scattering as random feedback, one would be able to achieve random parametric oscillation with a much wider wavelength tuning range. In that case, an FBG is no longer required: since Rayleigh scattering is proportional to $1/\lambda^4$, random parametric oscillation would preferably start at the shorter wavelength satisfying phase-matching (signal), while longer wavelengths satisfying phase-matching (idler) would not have enough feedback to oscillate. Thus, any non-selective mirror would suffice to form a half-open cavity, where wide wavelength tuning could be implemented by altering the crystal properties as in conventional OPO technology.

It should be mentioned that a novel concept of a piecewise distributed random cavity has been introduced and experimentally evaluated, which is by no means restricted to MI gain. Certainly, any kind of amplification mechanism combined with piecewise distributed feedback can be explored for the development of other types of random optical oscillators. We anticipate that the piecewise cavity will be investigated in the context of pulsed random fibre lasers, in which optical gain is given by stimulated emission of radiation rather than parametric conversion.

We demonstrated that the R-OPO might also find applications in the field of ultra-fast optics. The FWM by-products that immediately rise with the onset of random parametric oscillation contribute to the sharpening of MI-induced oscillations. In such a way, the R-OPO output is composed of picosecond-pulses train, readily available at arbitrary repetition rates. By filtering the oscillation line, the pulse train vanishes, and the output becomes a 55% narrower and sharper version of the pump pulses, but governed by random parametric oscillation.

## Data availability
The data that support the findings of this study are publicly available at https://zenodo.org/record/8377807.

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

## Acknowledgements

The authors would like to thank for the assistance of OFS-optics, specially Paul S. Westbrook, Zhou Shi and Ping Lu, who directly contributed to the wavelength hop-free results by loaning the specially designed enhanced-Rayleigh fibre. This work was supported by the Canada Research Chairs (950-231352) and Natural Sciences and Engineering Research Council of Canada (RGPIN-2020-06302, 06302/DGDND/2020).

## Author contributions

P.T. performed experimental studies, conducted numerical calculations and wrote the manuscript. P.T., J.P.v.d.W., Y.W., L.C. and X.B. contributed to the analysis.

## Competing interests

The authors declare no competing interests.
