## [Peer Review File · Nature Communications]

REVIEWER COMMENTS

Reviewer #1 (Remarks to the Author):

The authors report on a random optical parametric oscillator (ROPO) that is based on a half-open cavity composed of a fiber Bragg grating and a 5-km-long optical single mode fiber. The authors show that this approach allows for optical parametric oscillation with extremely weak feedback from Rayleigh scattering occurring in repetition-rate-matched zones of the long optical fiber. The gain is provided by modulation instability (MI).

The idea is original, and this manuscript describes to the best of my knowledge the first demonstration of the above concept.

The data are technically sound and analyzed carefully with solid support by theoretical models. The experimental observations are discussed and interpreted accurately. There are some missing details, and the presentation is partially below the journal's standard (comments later).

The author's concept does at the same time yield a multitude of weaknesses that might strongly hamper the practical impact of the demonstrated technology. Therefore, despite the originality of the concept, the significance of the approach and results in the current stage seems moderate to me.

The points I am referring to are in particular:

- There is a dilemma regarding the system's stability. The authors claim that a stabilization mechanism is not needed, which they support by figure 5. While the ROPO does continuously operate with about 5% rms, it also shows spectral jumps on the few-seconds-timescale. The observed mode hops would result in a net-broadening of the linewidth. From an application point of view this may be problematic. However, a FBG has already been used to stabilize the emission wavelength on one side of the cavity and the Rayleigh feedback can intrinsically not be easily stabilized.

- Spectral tunability. The great advantage of OPO technology is the nearly unlimited spectral tunability. The demonstrated ROPO surrenders this advantage by employing the FBG as end mirror.

- The ROPOs wavelength is such close to the pump laser that for practical reasons tuning the latter might be a favorable approach.

- The system does only work in a somewhat narrow operating range in terms of pump power, pulse duration, and repetition rate, owing to strong side effects.

- The achieved power conversion efficiency is extremely low ($1e-4$ according to figure 3, see comment on this below).

- Single-frequency or narrow linewidth operation seems out of reach due to very strong MI and FWM.

- Repetition rate tunability. The authors demonstrate successful repetition rate tunability. But, presumably, tuning is associated with slight spectral changes (not shown in the manuscript, but sufficiently clear from the data and description – please correct me if this is wrong). More importantly, while continuous tuning is possible, on an absolute scale the range of available repetition rates seems limited to few hundreds of kHz to a few MHz, which should also be considered a constrain when compared to “classical” synchronously pumped OPOs.

Again, I would like to emphasize that these practical limitations would hamper the applicability of the source to any real-world problem, but the fact remains that it is an original demonstration of a new concept.

There are some additional points that the authors may consider:

- In lines 31/32 the authors claim that in case of Q-switched or mode-locked lasers the repetition rate of the OPO is fixed and determined by the pump laser. The message is clear, but it should be pointed out that this is only the case for synchronously pumped OPOs (SPOPOs). Hence, it applies certainly more to mode-locked than to Q-switched pump lasers.

- Repetition rate tunable OPOs are of course available in the nanosecond regime pumped by Q-switched lasers, but they lack mutual pulse coherence. This should be mentioned. It would be interesting to know if mutual pulse coherence is given for the ROPO. I believe this is the case. If this is correct, the authors could use this to distinguish their system from typical nanosecond OPOs in this regard.

- The authors should comment on the efficiency of the system. From figure 3 an efficiency of approximately $1e-4$ can be inferred, but it is unclear whether this refers to a single spectral line or the entire converted power. I could not find information about the efficiency in the manuscript.

- In line 237 the authors emphasize the system's robustness against cavity length fluctuations. It is unclear to me which physical property of the system is meant by "cavity length" here.

- The full setup picture including the pump section would be helpful (may be placed in SI).

- At which port of the system are the data taken that are displayed in figures 2 – 5? This needs to be mentioned in the manuscript.

Reviewer #2 (Remarks to the Author):

The authors present an optical parametric oscillator system, which is based on random feedback, governed by Rayleigh scattering in a 5.25-km long single-mode fiber. Nonlinear gain is achieved by modulation instabilities (MI), which essentially represents parametric gain by phased-matched four-wave mixing. The random feedback is provided continuously from every position within the feedback fiber by weak backreflections due to intrinsic refractive index impurities in the fiber. Together with a fiber Bragg grating on one end facet the fiber forms a half-open cavity. Pulsed pumping on the nanosecond timescale is realized using a semiconductor laser in combination with two electro-optic modulators. The fact that backreflections occur at every arbitrary position in the fiber allows to operate the random optical parametric oscillator (R OPO) system at arbitrary pulse repetition rates. The authors experimentally demonstrate optical parametric oscillation in this system, characterize the threshold pump power level, demonstrate the formation of a frequency comb enabled by four-wave mixing, and quantify output power and wavelength stability.

First of all, I would like to point out that I enjoyed reading this paper and that I like the idea of transferring the random feedback concept from laser sources to a parametric light source. To the best of my knowledge, the concept presented in this work is novel and has never been demonstrated before. The paper is well organized and well written, and all the necessary details are provided. The authors explain the underlying physical principles in large detail. Furthermore, the experimental findings are convincing and consistent. Especially, the simulations support the measured data even further, and therefore, add to the overall quality of the paper. Also, I appreciate that the authors present their findings in an honest way without trying to conceal any limitations of the system.

However, from a pure application point of view, this system exhibits several limitations and, in my opinion, does not offer a key performance feature, which would render the R-OPO superior to existing light sources. In the following, these aspects are addressed in more detail:

(1) One main motivation of employing OPO systems is that they offer wavelength tunability, especially within spectral ranges that are not easily accessible with lasers. However, the output wavelength of the presented R OPO is determined by the spectral reflection profile of the fiber Bragg grating, which has to be tuned such that it coincides with the highest MI gain. Therefore, this system does not support wavelength tunability as, e.g., conventional fiber-feedback OPO (FFOPO) systems do by adjusting the cavity mismatch accordingly. So, in a sense, getting rid of cavity length control simplifies the system, but also prevents wavelength tunability.

(2) Furthermore, the R OPO output wavelength is intrinsically located close to the pump wavelength, as given by the MI gain profile. In this case the pump wavelength is 1549.05 nm and the signal output is located around 1550.0 nm. From an application standpoint I do not see any benefit from this system as a novel type of light source. Simple temperature tuning of the semiconductor pump diode might shift the wavelength to the same output.

(3) The next point concerns the power and wavelength stability. The authors point out, that compared to random lasers wavelength hopping occurs on a drastically reduced rate (line 222-223). Nevertheless, the wavelength stability does not seem to be feasible for applications that would otherwise benefit from narrowband laser pulses, such as spectroscopy or sensing applications.

(4) Apparently, Rayleigh scattering in the fiber, and therefore, the optical feedback, is susceptible to temperature and strain (line 209-212), which leads to wavelength hopping as demonstrated in Figure 5. This demands for temperature stabilization of the fiber. Therefore, the R-OPO does not improve/overcome the need for temperature stabilization. Actually, conventional FFOPO systems can be designed using gain crystals with fan-out poling gratings, such that phase matching does not rely on temperature tuning. As for the R-OPO, the cavity, i.e., mostly the feedback fiber, has to be temperature stabilized.

(5) The authors emphasize the arbitrarily tunable pulse repetition rate of their R OPO system. In principle, I agree that having the repetition rate available as an independent control parameter is desirable. However, as the authors describe from line 102 on, the repetition rate seems to be a necessary control parameter to precisely balance cavity losses and MI gain in order to enable OPO operation in the first place. Thus, the repetition rate might be arbitrarily tunable, but only within a relatively narrow range.

(6) As demonstrated in Figure 3, the pump power level has to be set precisely within a narrow range in order to, on the one hand, overcome the oscillation threshold, and, on the other hand, to avoid reduced MI gain by increasing SPM. This seems to make the system quite inflexible. Apart from that, the low absolute output power level of $\sim 30 \mu\text{W}$ would require post-amplification for most applications. This would add complexity and cost.

(7) In general, the system seems to exhibit a small parameter space, in which OPO can be sustained (pump power, repetition rate, temperature/strain of the fiber). Therefore, the aforementioned points

raise the question if the presented light source offers any significant advantage(s) over existing light sources, that would make it attractive for any kind of applications.

All in all, this work transfers the concept of random feedback from lasers to optical parametric oscillators, which, to the best of my knowledge, has not been demonstrated before. The experimental results are presented well and in an organized fashion. Simulations support the experimental findings.

Nevertheless, the presented light source seems to have no obvious advantage over existing light sources in terms of performance. The lack of apparent applications at this stage of the R-OPO concept might prevent the paper from generating immediate high impact, which of course remains to be assessed and decided by the editorial board. In my opinion, further work and development on this concept might pave the way towards a new (feasible) subclass of parametric light sources, which are based on random feedback. Therefore, I recommend this paper for publication in Nature Communications, given the following minor revisions are addressed:

- I would suggest, that a more detailed comparison between the R-OPO system and existing parametric light sources should be added to the manuscript. In particular, addressing the aspects listed above and putting them into context with other parametric light sources would strengthen the paper even further. In this context, potential for improvements could be stated as well (wavelength stabilization, output power scaling, potential concepts for introducing wavelength tunability, etc.).
- The visual appearance of the figures could be improved. In particular, the font sizes across the figures should be kept consistent. This would add to the quality of the figures.
- Figure S2 in the Supplementary Material shows the exceptional agreement between experiment and simulation. Personally, I would add these data to Figure 3 in the main manuscript, as I think it might strengthen the message of Figure 3.

I hope this review helps the editorial board during their decision process as well as the authors to further improve their manuscript.

Review : NCOMMS-23-07746-A

Title: A Random Optical Parametric Oscillator

Authors: Pedro Tovar, Jean Pierre von der Weid, Yuan Wang, Liang Chen and Xiaoyi Bao

AUTHORS' GENERAL COMMENTS:

We would like to thank the editor and reviewers for the valuable comments and suggestions received. As mentioned by both reviewers and the editor, despite the originality of the work, which transfers the random feedback concept from laser sources to optical parametric oscillators, a number of weaknesses that comes with the concept of random OPO (R-OPO) might hamper the direct application of the work. Although our goal with the initial submission was to report the physics of the first random OPO, including experimental evidence of a new phenomenon which would likely trigger the interest of research community in the field of complex systems in the same way that random lasers did, we agree that there were weaknesses, which could compromise its direct implementation in practical applications.

We put a lot of effort into overcoming these weaknesses, and we now provide a highly improved version of the manuscript. A major revision was implemented, with new experiments and new discussions. Four new experimental results are included in the new version (Fig. 4c, Fig. 6c-d, Fig 7b and Fig 7c-d), as well as discussions detailing the importance of these new results for practical utilisation of the R-OPO. We truly believe that the new results and discussions are proof of the high potential of the R-OPO for practical applications.

Together with the new version of the manuscript we are sending a marked version, highlighting the major changes from the initial submission. Below we provide a detailed response to the reviewers' comments.

REVIEWER #1:

The authors report on a random optical parametric oscillator (ROPO) that is based on a half-open cavity composed of a fiber Bragg grating and a 5-km-long optical single mode fiber. The authors show that this approach allows for optical parametric oscillation with extremely weak feedback from Rayleigh scattering occurring in repetition-rate-matched zones of the long optical fiber. The gain is provided by modulation instability (MI).

The idea is original, and this manuscript describes to the best of my knowledge the first demonstration of the above concept.

The data are technically sound and analyzed carefully with solid support by theoretical models. The experimental observations are discussed and interpreted accurately.

Authors' Response: We would like to thank the reviewer for the comments and for the detailed review. The reviewer raised several key points that sparked extensive discussions within our group, ultimately resulting in an enhanced version of the manuscript. We express our sincere gratitude for the valuable comments. Next, we carefully discuss the points raised by the reviewer.

There are some missing details, and the presentation is partially below the journal's standard (comments later).

The author's concept does at the same time yield a multitude of weaknesses that might strongly hamper the practical impact of the demonstrated technology. Therefore, despite the originality of the concept, the significance of the approach and results in the current stage seems moderate to me.

The points I am referring to are in particular:

Comment 1.1) There is a dilemma regarding the system's stability. The authors claim that a stabilization mechanism is not needed, which they support by figure 5. While the ROPO does continuously operate with about 5% rms, it also shows spectral jumps on the few-seconds-timescale. The observed mode hops would result in a net-broadening of the linewidth. From an application point of view this may be problematic. However, a FBG has already been used to stabilize the emission wavelength on one side of the cavity and the Rayleigh feedback can intrinsically not be easily stabilized.

Response to comment 1.1) We agree with the reviewer's analysis. The result shown in Fig. 6a-b (Fig. 5 in previous version) was obtained with a repetition rate of 1 MHz, thus corresponding to a feedback from 50 repetition-rate-matched zones along the 5.25 km fibre. Each of these sections has about 0.5 m, and neighboring sections are separated by about 100 m. Due to the inherent randomness of the refractive index in single-mode fibres, the backscattered spectrum of each section is random (see Supplementary Equation (2)). The effect of such spectral randomness is often observed in the time-domain when measuring Φ -OTDR traces, where the backscattered light from every location gives a temporal profile with random intensity fluctuations, so the backscattered spectrum possesses the same random nature, full of randomly located spectral spikes [28]. Random parametric oscillation initiates at the highest spectral reflectivity peak resulting from the coherent sum of the Rayleigh backscattered spectra from all 50 sections (see Supplementary Fig. 1). The backscattered Rayleigh pattern from each section is fairly stable, which is the reason why Φ -OTDR works for temperature/strain sensing. However, the stability is compromised when considering tens of sections separated by a hundred metres each. This, as correctly described by the reviewer, may be problematic for practical applications. Below, we describe the physical origin of this problem and the implemented solution.

The spectral reflectivity profile of a single fibre section depends both on the random distribution of the refractive index of the section, and on the optical phase

propagation along the section. In other words, since the reflected spectrum is given by the interference of light backscattered over the section, if the phase difference between light backscattered at the beginning and at the end of the section is stable, then the reflected spectrum would be stable too. Such a phase difference depends on the refractive index, n , wavenumber, k_0 , and section length, d :

$$\Delta\phi_d = n(T, \varepsilon) k_0 d$$

Since d is small (0.5 m), the phase difference is fairly robust against small temperature/strain fluctuations. Now considering two sections, the effective reflectivity spectrum is given by the interference of light backscattered from both sections. Even though $\Delta\phi_d$ is stable for each section analysed individually, the interference between two sections separated by a distance D depends on the phase propagation along D :

$$\Delta\phi_D = n(T, \varepsilon) k_0 D$$

Since D is much larger than d ($D \approx 100$ m), small variations of the refractive index along D , mostly caused by environmental perturbations, are multiplied by a large quantity, thus strongly impacting $\Delta\phi_D$. Hence, the resulting backscattering spectral profile fluctuates with environmental noise. Clearly, this effect is magnified when the effective reflectivity comes from the interference between tens of sections separated by a hundred metres each. Therefore, the spectral location of the highest reflectivity peak fluctuates with environmental noise, and hence shifting the parametric oscillation wavelength (mode hopping).

Even though this happens in a time scale of a few seconds, which is already far more stable than random fibre lasers, it leads to a net-broadening of the linewidth. A solution to this problem is to reduce the number of sections to one, aiming to achieve the same stability of Φ -OTDR systems. Obviously, this comes with the penalty of higher loss, since more sections translates into more backscattered power and a lower threshold for parametric oscillation. To mitigate this issue we cascaded the 5.2 km SMF with a 1 km-long enhanced-Rayleigh fibre, which provides about 16 dB extra backscattering light compared to standard SMF. Therefore, one section of the enhanced-Rayleigh fibre corresponds to ~ 40 sections of SMF in terms of backscattered power, but with a stable backscattered spectrum. By tuning the repetition rate of pump pulses to select only one repetition-rate-matched zone in the enhanced-Rayleigh fibre we were able to observe random parametric oscillation without wavelength hops, solving the net-broadening problem. This new result is included in the manuscript in Fig. 6c-d, and it also shows a better peak power stability compared to the case with 50 effective Rayleigh sections. The result was measured with the enhanced-Rayleigh fibre simply laying on top of a table with a lot of lab equipment turned on (with fan-coolers, i.e., vibration noise), which was not enough to disturb the emission wavelength. Yet, clearly, as the temperature/strain in the addressed section changes, a wavelength shift would be observed. This can be either solved through stabilizing the enhanced-Rayleigh fibre (e.g., put in a thermally isolated sound-proof box), or explored as a temperature/strain sensor with high spatial resolution and high signal-to-noise ratio. The high SNR is expected because one would

measure the wavelength drift of a strong oscillating light, not from a weak backscattering light as in most fibre-sensors.

The discussion above has been included in the new version of the manuscript in lines 249-292.

Comment 1.2) Spectral tunability. The great advantage of OPO technology is the nearly unlimited spectral tunability. The demonstrated ROPO surrenders this advantage by employing the FBG as end mirror.

Response to comment 1.2) Indeed, conventional OPO technology is known for its wide spectral tunability. The R-OPO demonstrated does have limitations in this sense. However, the main limitation in our case is not the FBG tunability, which can be tuned over several nanometres, but rather the tunability of the spectral range of MI sideband (parametric gain spectrum). The separation between the first MI sideband and the pumping wavelength depends on the pump power, β_2 (GVD parameter) and γ (non-linear parameter). Thus, without changing the fibre, i.e., with β_2 and γ fixed, the wavelength tuning depends mostly on the pump power used. For instance, in Fig. 2b we can see a small shift of the peak MI gain when increasing the power from 320 to 345 mW. Choosing a fibre with different parameters β_2 and γ would result in parametric oscillation at different wavelengths. Unfortunately, changing the fibre is not a practical solution for wavelength tuning. Another option would be to set the FBG centre wavelength to a higher order sideband of MI. We experimentally verified this alternative, and we observed the onset of parametric oscillation up to the third order sideband of MI, thus enabling a tunability of about 6 nm (from negative 3rd order sideband to positive 3rd order sideband). Although this tuning range is wider than most semiconductor lasers, it is much smaller compared to conventional OPO.

However, the concept of random parametric oscillation goes far beyond the setup explored in this paper. Different parametric amplification mechanisms other than MI could be used for parametric amplification. For instance, by exploring the $\chi^{(2)}$ non-linearity of crystals for parametric amplification, and Rayleigh scattering as random feedback, one would be able to achieve random parametric oscillation with a much wider tuning range. In that case, an FBG is no longer required: since Rayleigh scattering is proportional to $1/\lambda^4$, random parametric oscillation would preferably start at the lower wavelength satisfying phase-matching (signal), while longer wavelengths satisfying phase-matching (idler) wouldn't have enough feedback to oscillate, i.e., Rayleigh scattering would naturally select the oscillation wavelength. Thus, any non-selective mirror would suffice to form a half-open cavity, where wide wavelength tuning could be implemented by altering the crystal properties as in conventional OPO technology.

From what is said above, we do believe that the concept of random optical parametric oscillation is worth of publication, as it extends to different configurations, allowing, for instance, wide wavelength tuning without stringent requirements of conventional OPOs. We included this discussion in the new version of the manuscript in lines 362-383.

Comment 1.3) The ROPOs wavelength is such close to the pump laser that for practical reasons tuning the latter might be a favorable approach.

Response to comment 1.3) The properties of light generated from random parametric oscillation are highly different from those of pump laser, so that simply tuning the latter might not be a viable solution depending on the application. For instance, the pump laser linewidth is about 8 MHz, while the R-OPO emit short coherent pulses, with a 3 dB-linewidth of 2.5 kHz. For wider wavelength tuning capabilities, see the reply '*Response to comment 1.2*'. And for the discussion of inter-pulse coherence, which is essential for coherent LiDAR applications, see the new section added to the manuscript, '*Coherence*', in line 293.

Comment 1.4) The system does only work in a somewhat narrow operating range in terms of pump power, pulse duration, and repetition rate, owing to strong side effects.

Response to comment 1.4) We agree with the reviewer that the pump power range is limited, and one may need post-amplification depending on the application. Fortunately, fibre amplifiers are conveniently available as off-the-shelf devices at 1550 nm. Regarding pulse duration and repetition rate, in the new version of the manuscript we show that the R-OPO offers wide tunability. The pulse repetition rate of the R-OPO could be tuned from 16.6 kHz to beyond 2000 kHz, while synchronously pumped OPOs (SPOPOs) cannot offer any tunability in repetition rate. In addition, a recent work reporting a novel Q-switched nanosecond laser discusses the interest in repetition-rate tunable light sources, and authors show a 10 ns pulse train with a linewidth of 279.43 MHz offering repetition rate tunability from 1 to 4 kHz (ref [1] below), a far smaller range compared to the R-OPO, thus proving its value in terms of repetition rate tunability.

Regarding the pulse width tunability, we showed that it ranges from 0.69 ns to 47.9 ns (see new results in Fig. 4c). A recent work focusing on the fabrication of an OPO with tunable pulse width [2] showed tunability of the signal and idler from 5.7–12.6 ns and 7.1–19.7 ns, respectively. The R-OPO not only offers a wider tuning range compared with [2], but also emits highly coherent pulses with a tunable repetition rate, thus being of great interest for coherent LiDAR applications.

[1] Y. Chen, Z. Bai, D. Jin, Y. Li, Y. Qi, J. Ding, B. Yan, K. Wang, Y. Wang, and Z. Lu, "Repetition rate tunable single-longitudinal-mode acoustic-optical Q-switched nanosecond laser," *Results in Physics*, vol. 46, p. 106318, 2023.

[2] Z. Bai, C. Zhao, J. Gao, Y. Chen, S. Li, Y. Li, T. Liu, X. Yan, Y. Wang, and Z. Lu, "Optical parametric oscillator with adjustable pulse width based on KTiOAsO₄," *Optical Materials*, vol. 136, p. 113506, 2023.

Comment 1.5) The achieved power conversion efficiency is extremely low (1e-4 according to figure 3, see comment on this below).

Response to comment 1.5) Please, see '*Response to comment 1.11*'.

Comment 1.6) Single-frequency or narrow linewidth operation seems out of reach due to very strong MI and FWM.

Response to comment 1.6) Indeed, the output optical signal measured at point *A* in Fig. 2a includes strong MI and FWM by-products. However, by simply filtering the R-OPO output as shown in Fig. 6d, a single frequency/narrow linewidth operation is achieved.

Comment 1.7) Repetition rate tunability. The authors demonstrate successful repetition rate tunability. But, presumably, tuning is associated with slight spectral changes (not shown in the manuscript, but sufficiently clear from the data and description – please correct me if this is wrong). More importantly, while continuous tuning is possible, on an absolute scale the range of available repetition rates seems limited to few hundreds of kHz to a few MHz, which should also be considered a constrain when compared to “classical” synchronously pumped OPOs.

Response to comment 1.7) The reviewer’s understanding is correct. By tuning the repetition rate, different repetition-rate-matched zones are selected along the SMF; since each section has a random reflectivity spectrum, once the repetition rate is changed then slight spectral changes would be observed. However, since even for a fixed repetition rate mode hopping is observed, as shown in Fig. 6b, the slight spectral changes caused by abrupt changes in the repetition rate turns out to be irrelevant. Yet, when adding the enhanced-Rayleigh fibre in the setup, with which a mode hop-free emission was obtained, slight changes in the repetition rate certainly affect the emission wavelength. It is important to note that these changes are still constrained to the FBG bandwidth, so that they don’t affect much the emission wavelength.

We agree with the reviewer regarding the limitation of the repetition rate tunability. We were able to successfully tune the repetition rate from 16.6 kHz up to more than 2000 kHz. Even though the tuning range is limited, it is much wider than that obtained in similar works (see ‘*Response to comment 1.4*’). Surely, we cannot compare the tunability herein obtained with that of classical OPOs pumped with pulses much longer than the cavity length (not SPOPO). In these cases, which are normally pumped by Q-switched lasers, since the pump pulses’ width is much longer than the cavity length, they rather act like a CW pump, where the OPO lacks inter-pulse coherence. In our work, the R-OPO enables tunable repetition rate while at the same time offering inter-pulse coherence, which is a significant advantage compared to other pulsed light sources.

Comment 1.8) Again, I would like to emphasize that these practical limitations would hamper the applicability of the source to any real-world problem, but the fact remains that it is an original demonstration of a new concept.

Response to comment 1.8) We believe that the new version of the manuscript along with this response letter have addressed the practical limitations mentioned above, and that the original demonstration of the concept is worth sharing in Nature Communications.

There are some additional points that the authors may consider:

Comment 1.9) In lines 31/32 the authors claim that in case of Q-switched or mode-locked lasers the repetition rate of the OPO is fixed and determined by the pump laser. The message is clear, but it should be pointed out that this is only the case for synchronously pumped OPOs (SPOPOs). Hence, it applies certainly more to mode-locked than to Q-switched pump lasers.

Response to comment 1.9) We thank the reviewer for the comment. Indeed, we were referring to SPOPOs when mentioning that the repetition rate is fixed and should match the cavity's round-trip time. We clarified this point in the new version of the manuscript in lines 46, 176 and 340.

Comment 1.10) Repetition rate tunable OPOs are of course available in the nanosecond regime pumped by Q-switched lasers, but they lack mutual pulse coherence. This should be mentioned. It would be interesting to know if mutual pulse coherence is given for the ROPO. I believe this is the case. If this is correct, the authors could use this to distinguish their system from typical nanosecond OPOs in this regard.

Response to comment 1.10) We largely appreciate the reviewer's comment, which triggered new experiments and discussions that highly improved the quality of the work. New measurements were performed to verify mutual-pulse coherence, and a new section was added to the paper: '*Coherence*'. There, we validate the reviewer's expectation and further discuss mutual-pulse coherence results. In addition, in the '*Discussion*' section we highlight the importance of a coherent pulsed source for LiDAR applications.

Comment 1.11) The authors should comment on the efficiency of the system. From figure 3 an efficiency of approximately $1e-4$ can be inferred, but it is unclear whether this refers to a single spectral line or the entire converted power. I could not find information about the efficiency in the manuscript.

Response to comment 1.11) From the result presented in Fig. 3c (new version of the manuscript), the system has an efficiency of 2.6×10^{-3} . We included this information in line 148. In addition, we clarify that the result shown in Fig. 3c corresponds to the measurement of optical power at the R-OPO oscillating line around 1550 nm (not the entire converted power), which was collected at the output of the setup (point *A* in Fig. 2a). This information is included in line 119.

Comment 1.12) In line 237 the authors emphasize the system's robustness against cavity length fluctuations. It is unclear to me which physical property of the system is meant by "cavity length" here.

Response to comment 1.12) Indeed, when discussing systems with random feedback the definition of 'cavity length' is unclear. We already modified the passage in line 237 (currently, in line 340) to avoid confusions. Our intention was to refer to the SMF where random parametric oscillation takes place, but we agree that using the expression 'cavity length' is not accurate. This has been fixed in the new version of the manuscript.

Comment 1.13) The full setup picture including the pump section would be helpful (may be placed in SI).

Response to comment 1.13) We thank the reviewer for the suggestion. The full setup picture has been included in Fig. 2a.

Comment 1.14) At which port of the system are the data taken that are displayed in figures 2 – 5? This needs to be mentioned in the manuscript.

Response to comment 1.14) When including the full setup picture, we identified 3 key points in Fig. 2a: points *A*, *B* and *C*. Most measurements were conducted at point *A*, and each time we introduced a new measurement result along the text, we mentioned the measurement point.

REVIEWER #2:

Reviewer #2 Comments: The authors present an optical parametric oscillator system, which is based on random feedback, governed by Rayleigh scattering in a 5.25-km long single-mode fiber. Nonlinear gain is achieved by modulation instabilities (MI), which essentially represents parametric gain by phased-matched four-wave mixing. The random feedback is provided continuously from every position within the feedback fiber by weak backreflections due to intrinsic refractive index impurities in the fiber. Together with a fiber Bragg grating on one end facet the fiber forms a half-open cavity. Pulsed pumping on the nanosecond timescale is realized using a semiconductor laser in combination with two electro-optic modulators. The fact that backreflections occur at every arbitrary position in the fiber allows to operate the random optical parametric oscillator (R OPO) system at arbitrary pulse repetition rates. The authors experimentally demonstrate optical parametric oscillation in this system, characterize the threshold pump power level, demonstrate the formation of a frequency comb enabled by four-wave mixing, and quantify output power and wavelength stability.

First of all, I would like to point out that I enjoyed reading this paper and that I like the idea of transferring the random feedback concept from laser sources to a parametric light source. To the best of my knowledge, the concept presented in this work is novel and has never been demonstrated before. The paper is well organized and well written, and all the necessary details are provided. The authors explain the underlying physical principles in large detail. Furthermore, the experimental findings are convincing and consistent. Especially, the simulations support the measured data even further, and therefore, add to the overall quality of the paper. Also, I appreciate that the authors present their findings in an honest way without trying to conceal any limitations of the system.

Authors' Response: We thank the reviewer for the kind comments, and we are happy to know that it was an enjoyable reading. Indeed, we believe that transferring the concept of random feedback from lasers to parametric oscillators is a novel approach with great potential in the field of parametric sources. Also, we would like to thank for the thorough review, which helped us to revise the

manuscript and reach a significantly improved version. Next, we carefully discuss the points raised by the reviewer.

However, from a pure application point of view, this system exhibits several limitations and, in my opinion, does not offer a key performance feature, which would render the R-OPO superior to existing light sources.

Authors' Response: We agree with the reviewer that the R-OPO has limitations, some of which were addressed and solved in the new version of the manuscript. The main purpose of this work is to put forward the concept of random optical parametric oscillation, which has been experimentally verified for the first time—here, through MI gain and Rayleigh scattering in SMF, although different parametric amplification mechanisms and/or random feedbacks could be chosen (see second paragraph of '*Response to comment 1.2*'). Even though the R-OPO has features that make it more attractive than conventional OPOs for some applications (see updated '*Discussion*' section), our goal was not to develop a parametric oscillator superior to all others. In the same way that the concept of random lasers lead to outstanding findings, e.g., the first experimental evidence of replica symmetry breaking [Ref 1 below, interestingly enough published in *Nat. Commun.*], which is a concept that granted the Nobel prize in physics to Giorgio Parisi in 2021, we believe that random optical parametric oscillators can not only lead to practical optical sources for specific applications, but also serve as a platform for the study of complex systems. Furthermore, random lasers do not outperform lasers in every aspect, but they find many practical applications as detailed in the following reviews [Ref 2, Ref 3]. Likewise, it is expected that the report of the first random OPO will pave the way for new studies of complex physical systems as well as novel applications.

[Ref 1] N. Ghofraniha, I. Viola, F. Di Maria, G. Barbarella, G. Gigli, L. Leuzzi, and C. Conti, "Experimental evidence of replica symmetry breaking in random lasers," *Nat. Commun.* 6, 6058 (2015).

[Ref 2] Dmitry V. Churkin, Srikanth Sugavanam, Ilya D. Vatnik, Zinan Wang, Evgenii V. Podivilov, Sergey A. Babin, Yunjiang Rao, and Sergei K. Turitsyn, "Recent advances in fundamentals and applications of random fiber lasers," *Adv. Opt. Photon.* 7, 516-569 (2015).

[Ref 3] Anderson S.L. Gomes, André L. Moura, Cid B. de Araújo, Ernesto P. Raposo, "Recent advances and applications of random lasers and random fiber lasers," *Progress in Quantum Electronics*, Volume 78, 100343, (2021).

In the following, these aspects are addressed in more detail:

Comment 2.1) One main motivation of employing OPO systems is that they offer wavelength tunability, especially within spectral ranges that are not easily accessible with lasers. However, the output wavelength of the presented R OPO is determined by the spectral reflection profile of the fiber Bragg grating, which has to be tuned such that it coincides with the highest MI gain. Therefore, this system does not support wavelength tunability as, e.g., conventional fiber-feedback OPO (FFOPO) systems do by adjusting the cavity mismatch accordingly. So, in a sense, getting rid of cavity length control simplifies the system, but also prevents wavelength tunability.

Response to comment 2.1) Indeed, getting rid of cavity length control simplifies the system, but it does not completely prevent wavelength tunability. As discussed in lines 368-370, by tuning the FBG to higher order MI sidebands, we achieved a wavelength tuning of about 6 nm. Certainly, this is a far smaller tuning range compared to conventional OPOs, but is a wider range compared to most semiconductor lasers. In addition, as discussed in *'Response to comment 1.2'*, different configuration of R-OPO could potentially lead to wide wavelength tuning capabilities while still in a simplified configuration. Certainly, this is a point for future validation, but the fact that the R-OPO evaluated in this work is robust against environmental perturbations is of great interest for applications where wide wavelength tunability is not required.

Comment 2.2) Furthermore, the R OPO output wavelength is intrinsically located close to the pump wavelength, as given by the MI gain profile. In this case the pump wavelength is 1549.05 nm and the signal output is located around 1550.0 nm. From an application standpoint I do not see any benefit from this system as a novel type of light source. Simple temperature tuning of the semiconductor pump diode might shift the wavelength to the same output.

Response to comment 2.2) The same comment has been made by Reviewer 1, so we address Reviewer 2 to *'Response to comment 1.3'* above.

Comment 2.3) The next point concerns the power and wavelength stability. The authors point out, that compared to random lasers wavelength hopping occurs on a drastically reduced rate (line 222-223). Nevertheless, the wavelength stability does not seem to be feasible for applications that would otherwise benefit from narrowband laser pulses, such as spectroscopy or sensing applications.

Response to comment 2.3) Once again, the same concern has been raised by Reviewer 1, so we address Reviewer 2 to *'Response to comment 1.1'* above.

Comment 2.4) Apparently, Rayleigh scattering in the fiber, and therefore, the optical feedback, is susceptible to temperature and strain (line 209-212), which leads to wavelength hopping as demonstrated in Figure 5. This demands for temperature stabilization of the fiber. Therefore, the R-OPO does not improve/overcome the need for temperature stabilization. Actually, conventional FFOPO systems can be designed using gain crystals with fan-out poling gratings, such that phase matching does not rely on temperature tuning. As for the R-OPO, the cavity, i.e., mostly the feedback fiber, has to be temperature stabilized.

Response to comment 2.4) This concern is also covered in *'Response to comment 1.1'* above.

Comment 2.5) The authors emphasize the arbitrarily tunable pulse repetition rate of their R OPO system. In principle, I agree that having the repetition rate available as an independent control parameter is desirable. However, as the authors describe from line 102 on, the repetition rate seems to be a necessary control parameter to precisely balance cavity losses and MI gain in order to enable OPO operation in the first place. Thus, the repetition rate might be arbitrarily tunable, but only within a relatively narrow range.

Response to comment 2.5) As the reviewer correctly mentioned, the repetition rate is a parameter that can be used to balance cavity losses and MI gain. As shown in Fig. 3d of the new version of the manuscript, the threshold power for parametric oscillation is indeed a function of the repetition rate. However, by increasing the pump power, and thus increasing MI gain (which could also be increased by using longer fibres), a wide repetition rate tuning range could be obtained. When using short pump pulses (see discussion in lines 200-219), with duration of 2 ns, the repetition rate could be tuned from 700 kHz beyond 2 MHz. For wider pulses, a wider tuning range was achieved, starting from about 40 kHz. See ‘*Response to comment 1.7*’ for further details on repetition rate tuning.

Comment 2.6) As demonstrated in Figure 3, the pump power level has to be set precisely within a narrow range in order to, on the one hand, overcome the oscillation threshold, and, on the other hand, to avoid reduced MI gain by increasing SPM. This seems to make the system quite inflexible. Apart from that, the low absolute output power level of $\sim 30 \mu\text{W}$ would require post-amplification for most applications. This would add complexity and cost.

Response to comment 2.6) Indeed, when pumping with 5 ns pulses at a repetition rate of 1 MHz, the operation range in terms of pump power is rather small. However, this range could be highly enlarged by changing the pulse width and repetition rate. As shown in Fig. 3d, lower oscillation thresholds are obtained for higher repetition rates; as the MI gain saturation is independent of the repetition rate, then a wider operating range is immediately obtained for higher repetition rates. In a similar way, the oscillation threshold is smaller for wider pulses, thus also offering wider operation range in terms of pump power. These aspects have been added to the manuscript in lines 183-190 and 200-219.

Regarding the absolute power level, indeed it would require post-amplification depending on the application. Fortunately, fibre amplifiers are conveniently available as off-the-shelf devices at 1550 nm, which, however adds cost to the system, we believe it does not represent a large increase in complexity.

Comment 2.7) In general, the system seems to exhibit a small parameter space, in which OPO can be sustained (pump power, repetition rate, temperature/strain of the fiber). Therefore, the aforementioned points raise the question if the presented light source offers any significant advantage(s) over existing light sources, that would make it attractive for any kind of applications.

Response to comment 2.7) The same comment has been made by Reviewer 1, so we address reviewer 2 to ‘*Response to comment 1.4*’ above. In addition, we added a discussion in lines 357-361 highlighting the importance of a highly coherent pulsed light source with tunable repetition rate and tunable pulse width for LiDAR applications.

Comment 2.8) All in all, this work transfers the concept of random feedback from lasers to optical parametric oscillators, which, to the best of my knowledge, has not

been demonstrated before. The experimental results are presented well and in an organized fashion. Simulations support the experimental findings.

Nevertheless, the presented light source seems to have no obvious advantage over existing light sources in terms of performance. The lack of apparent applications at this stage of the R-OPO concept might prevent the paper from generating immediate high impact, which of course remains to be assessed and decided by the editorial board. In my opinion, further work and development on this concept might pave the way towards a new (feasible) subclass of parametric light sources, which are based on random feedback. Therefore, I recommend this paper for publication in Nature Communications, given the following minor revisions are addressed:

Response to comment 2.8) We agree with the reviewer that the concept of R-OPO paves the way towards a new subclass of parametric light sources. In addition, regarding the reviewer's concern of a direct application of the R-OPO, we point out two practical applications. First, as detailed in the second paragraph of the '*Discussion*' section, given the flexibility of the R-OPO in generating optical pulses with tunable repetition rates and pulse widths, coupled with its highly coherent emission, it emerges as a light source of considerable interest for LiDAR applications. And second, as described in lines 288-292, the repetition-rate-matched zone in the Rayleigh-enhanced fibre can serve as a distributed temperature/strain sensor with high SNR, opening a new class of OPO-based distributed sensors. Thus, we believe the new version of the manuscript describes a new light source with rich physics to be explored and readily implemented.

Comment 2.9) I would suggest, that a more detailed comparison between the R-OPO system and existing parametric light sources should be added to the manuscript. In particular, addressing the aspects listed above and putting them into context with other parametric light sources would strengthen the paper even further. In this context, potential for improvements could be stated as well (wavelength stabilization, output power scaling, potential concepts for introducing wavelength tunability, etc.).

Response to comment 2.9) We thank the reviewer for the suggestion. New references were included to provide a more detailed comparison between the R-OPO and other light sources (see all references added in the 'Reference' section in the marked version). For instance, in lines 354-357 we compare the pulse width tunability of the R-OPO with that of a recently proposed OPO which also allows tunable pulse width (ref [31]). In lines 351-354, we compared this work with that of reference [42], which describes the development of a nanosecond pulsed laser source with tunable repetition rate designed for LiDAR applications. In terms of wavelength tuning, we compared this work to a recent publication of a widely tunable fibre-feedback OPO - see lines 371-373. We believe these comparisons put this work into context, clearly showing the advantages of the R-OPO.

In addition, since the new version of the manuscript already includes a solution for wavelength stabilization (see new result included in Fig. 6d), we included in lines 374-383 potential concepts for improving wavelength tunability, which we plan to investigate in a future work.

Comment 2.10) The visual appearance of the figures could be improved. In particular, the font sizes across the figures should be kept consistent. This would add to the quality of the figures.

Response to comment 2.10) We modified all figures to keep consistent font size and font family. In addition, all figures were included in vector-graphics form (.eps format), enhancing the quality of the manuscript. We also followed Nature Communication guidelines regarding the usage of colors in scientific graphs according to [Ref 1 below].

[1] Crameri, F., Shephard, G.E. & Heron, P.J. “The misuse of colour in science communication.” Nat Commun 11, 5444 (2020).

Comment 2.11) Figure S2 in the Supplementary Material shows the exceptional agreement between experiment and simulation. Personally, I would add these data to Figure 3 in the main manuscript, as I think it might strengthen the message of Figure 3.

Response to comment 2.11) We thank the reviewer for the suggestion. It was implemented accordingly. See Fig. 3a-b in the new version of the manuscript.

REVIEWERS' COMMENTS

Reviewer #1 (Remarks to the Author):

I appreciate the authors time and dedication in answering my comments. All concerns are addressed thoroughly and I agree with the authors statements.

Overall, the authors have greatly improved the manuscript. In particular, two new major experimental additions have been made. One relates to the coherence of the emitted pulse train. The other addition demonstrates stability improvement by incorporating a enhanced-Rayleigh fiber into the feedback. Both measurements add value to the manuscript. I appreciate that the authors state both the benefits and the limitations associated with the enhanced-Rayleigh fiber. Furthermore the time-domain analysis section reads more clearly with modified Fig. 4. The figure quality has been substantially improved. The (sub-) figure rearrangements that have been made strengthen the paper. There remain a few weak points regarding the applicability of this source, but given the novelty of the approach, as well as the clear analysis and the insight into the physics of this interesting system, I recommend this manuscript for publication.

Reviewer #2 (Remarks to the Author):

The authors carefully addressed the concerns raised in my initial review, as well as those of the other reviewer. Major revisions have been carried out including additional experiments and detailed discussions, which were pointed out in the initial review. Clearly, the concept of ROPO may still have weaknesses in performance compared to other technologies. However, as the authors point out, the concept of ROPO also holds the potential for practical applications, given that further research will be carried out. Therefore, this paper will (hopefully) stimulate the interest in the community and lead to further advances of this concept. In my opinion, the changes made increased the overall quality of the paper accordingly. Therefore, I recommend this paper for publication in Nature Communications as it is in its current form.